# Energy Consumption and Carbon Dioxide Production Optimization in an Educational Building Using the Supported Vector Machine and Ant Colony System

**Wongchai Anupong** [1,*], **Iskandar Muda** [2] , **Sabah Auda AbdulAmeer** [3], **Ibrahim H. Al-Kharsan** [4],
**Aníbal Alviz-Meza** [5,*] and **Yulineth Cárdenas-Escrocia** [6]

1 Department of Agricultural Economy and Development, Faculty of Agriculture, Chiang Mai University, Chiang Mai 52000, Thailand
2 Department of Doctoral Program, Faculty Economic and Business, Universitas Sumatera Utara, Medan 20222, Indonesia
3 Department of Mechanical Engineering, Ahl Al Bayt University, Kerbala 56001, Iraq
4 Computer Technical Engineering Department, College of Technical Engineering, The Islamic University, Najaf 54001, Iraq
5 Grupo de Investigación en Deterioro de Materiales, Transición Energética y Ciencia de Datos DANT3, Facultad de Ingenieria y Urbanismo, Universidad Señor de Sipán, Km 5 Via Pimentel, Chiclayo 14001, Peru
6 GIOPEN, Energy Optimization Research Group, Energy Department, Universidad de la Costa (CUC), Cl. 58 ##55–66, Barranquilla 080016, Atlántico, Colombia
* Correspondence: anupong.w@cmu.ac.th or anupong.wongchai@yahoo.com (W.A.); alvizanibal@crece.uss.edu.pe (A.A.-M.)

**Abstract:** Buildings account for sixty percent of the world's total annual energy consumption; therefore, it is essential to find ways to reduce the amount of energy used in this sector. The road administration organization in Jakarta, Indonesia, utilized a questionnaire as well as the insights of industry experts to determine the most effective energy optimization parameters. It was decided to select variables such as the wall and ceiling materials, the number and type of windows, and the wall and ceiling insulation thickness. Several different modes were evaluated using the Design-Builder software. Training the data with a supported vector machine (SVM) revealed the relationship between the inputs and the two critical outputs, namely the amount of energy consumption and $CO_2$ production, and the ant colony algorithm was used for optimization. According to the findings, the ratio of the north and east windows to the wall in one direction is 70 percent, while the ratio of the south window to the wall in the same direction ranges from 35 to 50 percent. When the ratio and percentage of the west window to the west wall is between 60 and 70 percent, the amount of produced energy and $CO_2$ is reduced to negligible levels.

**Keywords:** building; energy optimization; ACS; SVM

## 1. Introduction

Energy consumption in buildings accounts for 60% of the world's annual energy consumption and 43% of the world's greenhouse gas emission [1–4]. Emissions of greenhouse gases, particularly $CO_2$, have a significant impact on both climate change and the warming of the planet [5]. Climate change and global warming cause many challenges to the world such as increases in urban floods [6–8]. The first step towards achieving smart buildings is energy consumption analysis and optimization. The growth of electric energy consumption and its dependence on polluting fossil fuels, especially in domestic and commercial uses, has led researchers to think of ways to control the amount of energy consumption in buildings [9]. Therefore, it is very important to provide solutions that can reduce energy consumption in this sector [10]. Compliance with the smallest details can have a great impact on reducing energy consumption in buildings; for example, the way the building is

oriented, the way the side spaces are located, and the improvement of insulation methods with the lowest cost can improve the energy efficiency of buildings, and thus by correcting the construction methods building design can achieve ideal design [11–13].

Most of the parameters of a building, including the thermal conductivity of the walls, the air permeability, the number of occupants, and the consumption of heating and cooling devices, are uncertain in nature, because they depend on changes in the weather and the behavior of the residents [14,15]. Saving energy consumption by paying attention to the placement of spaces in the plan based on matching the pattern of space occupation with the solar cycle, choosing the shell suitable for the climate and environmental conditions, setting the dimensions of the opening according to the received radiation and heat loss resulting from it, as well as the types of opening and glass are effective in reducing building energy consumption [16]. Carrying out proper planning in the field of energy consumption and management in smart educational buildings has made it possible to reduce more than 30% of annual energy consumption [17]. A multi-objective optimization method determines the building's exterior insulation thickness based on environmental and economic factors. Buildings with white or near-white exteriors, medium-capacity and thermal-resistant building materials, and tiny windows with awnings have lower inside air temperatures throughout the day [18]. However, buildings with dark exterior walls or large windows without awnings will have warmer indoor air than outside air during the day. Therefore, the importance of ventilation in changing the indoor air of a building depends on the external surface of its walls, as well as the size of the windows and the quality of the canopy. Using optimization, a multi-objective optimization model is presented to improve building energy performance [12].

In recent decades, machine learning methods have been used to model many scientific problems, and the results obtained from these methods have a significant compatibility with the measured values in the scale of real data [13,19–21]. The Support Vector Machine (SVM) method is also one of the machine learning methods that was introduced in the 1990s [12,13,18]. Despite the fact that the SVM method is a relatively new method, its wide use in the modeling of phenomena has been established and the results have been satisfactory [22,23].

In this study, the SVM method was used to predict the energy consumption of educational facilities. This article's primary objectives are to present and develop a reliable model for predicting building energy consumption using the SVM method and to optimize the effective parameter in energy usage in educational buildings using the ant colony system (ACS). First, the building's effective parameters are evaluated. Effective parameters are parameters whose value changes have noticeable effects on the energy consumption of the building. Then, an educational building in Jakarta, Indonesia is modeled using DesignBuilder (V5.5.2.007) tools and the effect of these parameters on the building's energy consumption and $CO_2$ is studied. As input data for the soft computing and optimization method, the building's physical characteristics, such as the window-to-wall ratio, were evaluated. By examining this topic, we attempt to answer the question of how the physical characteristics of a school building affect energy consumption and $CO_2$ emissions.

## 2. Materials and Methods

### 2.1. Case Study

This study chose the road administration organization building in Jakarta as a case study (Figure 1). The environmental data for modeling are related to Jakarta, Indonesia. A summary of Jakarta's climate conditions is presented in Table 1. In the present study an educational building in Jakarta was modelled based on these environmental conditions.

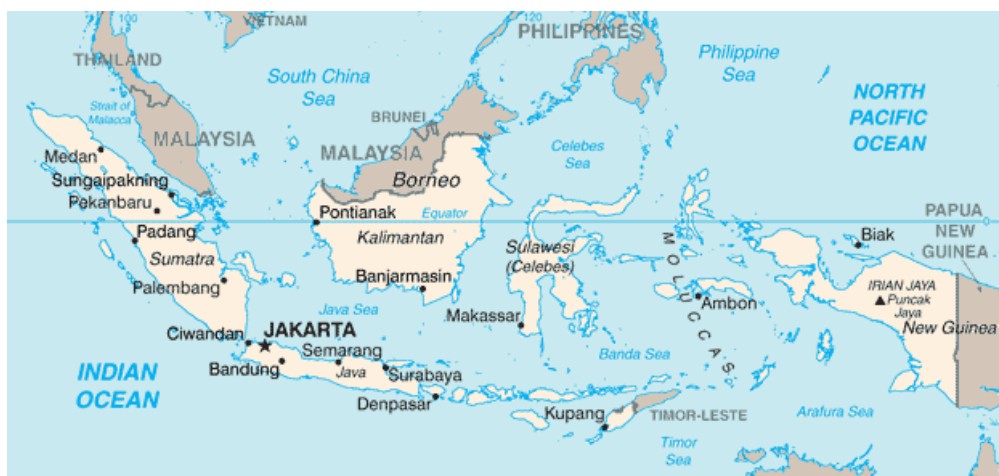

**Figure 1.** Jakarta, Indonesia's location in the South Pacific Ocean (https://cdn.worlddata.info/pics/countrymaps/IDN.png (accessed on 10 May 2017)).

**Table 1.** Summary of climate conditions in Jakarta.

| Annual Values | Jakarta, Sumatra, Indonesia |
|---|---|
| Daytime maximum temperature | 31.70 °C |
| Daily low temperature | 23.60 °C |
| Water temperature | 28.20 °C |
| Humidity | 83% |
| Precipitation | 2584 mm |
| Rain days | 152.4 days |
| Hours of sunshine | 1789 h |

### 2.2. DesignBuilder Model Validation

To validate the model of the educational building according to Figure 2, it was simulated in DesignBuilder.

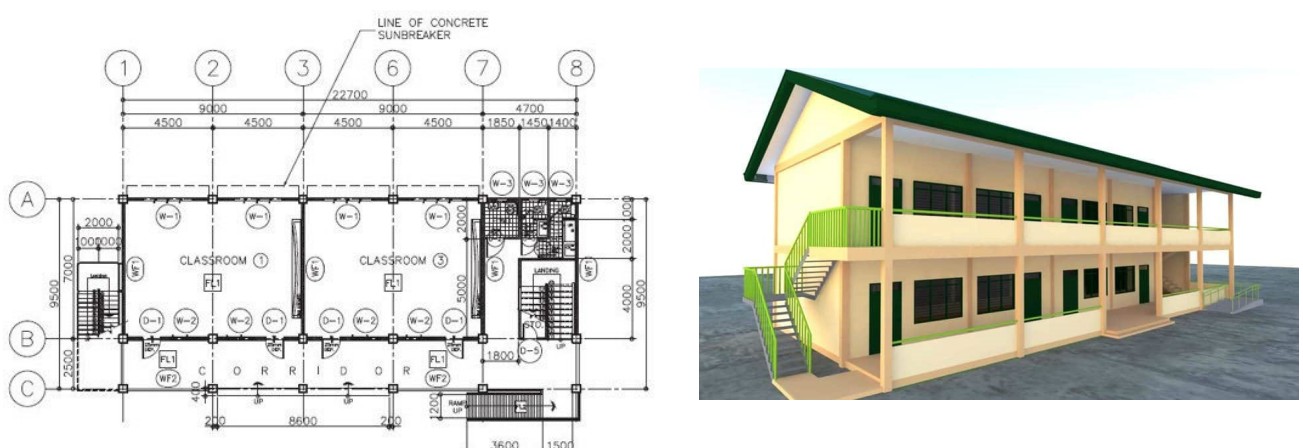

**Figure 2.** The modeled building characteristics and scheme.

The temperature of a classroom was evaluated in order to validate the model. It is assumed in the models that the surrounding air has a constant temperature (by the heat guard). Establishing the conditions is the initial step in the procedure for validating the

DesignBuilder results. This is relevant for DesignBuilder's adaptation of the associated graphs of practical testing and simulation. According to the RMSE, the statistical analysis of the numerical and experimental results of temperature reveals that the model's performance is satisfactory (RMSE = 1.08). In capturing the experiments accurately, the models meet all objectives (Figure 3).

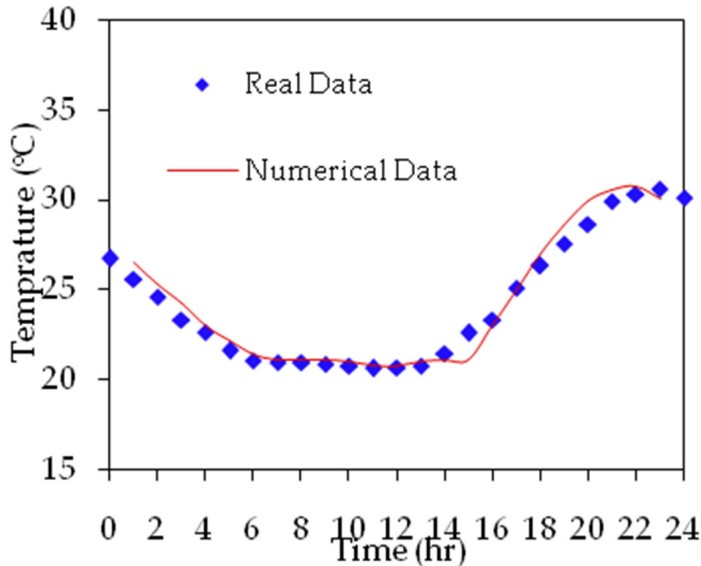

**Figure 3.** A comparison of numerical outcomes and real data.

To minimize energy usage and greenhouse gases emissions for low-energy buildings, it is required to first examine the desired building's existing energy consumption and then compare it with the numerical results. For this purpose, monthly electricity bills for the building in 2020 were created and adjusted for comparison with the numerical data.

The hottest months of the year (June and July) have the highest electricity consumption, while the peak electricity load is 409,555 kilowatt hours and the lowest electricity demand is 49,868 kilowatt hours, according to studies regarding electricity bills.

In the existing state, the wall material is brick, the roof material is beam, the cooling system is split, and the window is made of wood with double-glazed glass. The aforementioned data were entered into the DesignBuilder software, and the heat transfer coefficient for each building component was calculated. To simulate and enter the details, the software's information was utilized. The details of the roof and wall of the desired building in DesignBuilder before optimization are shown in Table 2. In this situation, the heat transmission coefficient of the roof is 1.456 W/m²k. The materials utilized in the construction of the external-to-interior brick wall are exterior brick and plaster and the heat transfer coefficient of the wall in this instance is 2.650 W/m²k.

Various energy-optimization techniques have been offered to minimize the building's energy consumption, including optimizing the window-to-wall ratio, wall and ceiling insulation, employing triple-glazed windows, and utilizing varied roof and wall detailing. In each optimization solution, the amount of cooling load reduction is computed, and the best state is then simulated by merging these solutions. In addition, Table 2 displays the roof and wall features of the intended structure in the DesignBuilder software's optimal mode. In the optimal mode, the materials used in the concrete slab roof are, from top to bottom, plaster, light concrete, and concrete; in this instance, the heat transfer coefficient of the roof is 1.520 W/m²k. The materials utilized in the concrete wall from the exterior to the interior of the building are light concrete and plaster and the heat transfer coefficient of the wall in this instance is 2.291 W/m²k.

The use of a brick wall compared to a concrete wall, a beam roof compared to a concrete slab roof, triple-paned glass compared to double-paned glass, the use of 40 mm polystyrene insulation compared to 35 mm insulation in a brick wall were investigated. In addition, the beam roof reduces the amount of cooling energy of the building and reduces the heat loss of the building.

**Table 2.** Characteristics regarding the materials utilized for the existing and optimal ceiling and interior and exterior walls.

| Building Elements | Materials | Heat Transfer Coefficient (W/m$^2$.k) | Thickness (mm) |
|---|---|---|---|
| **Existing** | | | |
| Reinforced concrete girder | Mosaic | | 22 |
| | Plaster | | 45 |
| | Waterproofing | | 45 |
| | Concrete | 1.456 | 105 |
| | Block | | 250 |
| | Plaster | | 45 |
| Brick wall | | 2.650 | 220 |
| **Optimal** | | | |
| Concrete Slab | Bituminous waterproofing | | 32 |
| | Plaster | | 25 |
| | Lightweight concrete | 1.520 | 40 |
| | Concrete | | 250 |
| | Plaster | | 20 |
| Concrete wall | Lightweight concrete | | 125 |
| | Plaster | 2.291 | 22 |
| Roof insulation | Polystyrene | 0.750 | 35 |
| | Polystyrene | 0.645 | 40 |
| Wall insulation | Polystyrene | 0.844 | 35 |
| | Polystyrene | 0.541 | 40 |
| Double-glazed glass | Two layers of glass | | 8 |
| | Argon gas | 1.562 | 3 |
| Window triple glass | Two layers of transparent glass | | 4 |
| | Low emissivity glass | 0.656 | 1 |
| | Krypton gas | | 15 |

All of the facts about the building, which was identified as successful in energy optimization in the previous section, are entered into this part of the software, and the software optimizes based on the least amount of energy consumed and the least amount of $CO_2$ emitted. The key element in this section is that the brick wall is compared to the concrete wall, the beam ceiling is compared to the concrete slab, triple- and double-glazed glass is compared to regular glass, and wall and ceiling insulation is 40 mm thick in optimization. The building energy is effective and generates reduced energy loss; therefore, it has been added as an effective choice in the DesignBuilder optimization. The window-to-wall percentages on the north, south, east, and west fronts, on the other hand, are defined as 15, 20, 45, and 60 percent, and the output of this component includes the total energy and $CO_2$.

### 2.3. Support Vector Machine (SVM)

Vapnik and Chapelle [24] developed the support vector machine (SVM), a two-tier classification approach incorporating observer learning. In this strategy, structural risk minimization (SRM) principles are employed to reduce the model error. Other methods (such as the method of artificial neural networks) employ experimental risk minimization (ERM) concepts.

Classification and regression problems requiring three or more categories are ideal candidates for SVR's use. Like many other machine learning techniques, this one has a two-step model-building process that consists of training and testing [25]. At the completion of the training process, the generalizability of the trained model is evaluated using test data. In practice, the support vector machine estimates the regression function using a series of linear functions. In this function, it is permissible to differ from the expected value (loss function). The ideal solution is then identified by applying structural risk reduction concepts to the loss-function-measured risk [26,27].

Solving the regression problem on a set of data in the form of $\{(x_1, y_1), \ldots (x_1, y_1) \epsilon R^m \ y \epsilon R\}$ is a linear function in the form of Equation (1) that can estimate the output values based on the inputs.

$$f(x) = \langle w.x \rangle + b \tag{1}$$

In the data set and Equation (1) $x$ is the input vector ($x \in R^m \ y$), $y$ is the output value ($y \in R^m$), w is the weight vector ($w \in R^m$), and b is the bias ($b \in R^m$)

The following optimization problem may be solved to acquire the control parameters of the optimal response function in SVM; that is, the weight and bias function. The loss function is used to obtain these control parameters.

$$\text{Minimize } \Phi(w, \zeta^*, \zeta) = \frac{\|w\|^2}{2} + C(\sum \zeta_i^* + \sum \zeta)$$
$$y_i - ((w \cdot x_i) + b) \leq \varepsilon + \zeta_i \tag{2}$$
$$\text{Subject to} \qquad ((w_1 x_i) + b) - y_i \leq \varepsilon + \zeta_i^* \quad \begin{array}{l} \zeta_i, \zeta_i \geq 0 \\ i = 1, 2, \ldots, 1 \end{array}$$

where $\| w \|^2$ is the weight vector norm and $\zeta_i^*$ and $\zeta$ are slack.

In accordance with Lagrange's theory, the optimization issue represented by Equation (3) may be recast as Lagrange's function, which is as follows:

$$L(\alpha^*, \alpha) = -\varepsilon \sum_{i=1}^{1} (\alpha_i^* + \alpha_i) +$$
$$\sum_{i=1}^{1} y_i(\alpha_i^* - \alpha_i) - \frac{1}{2} \sum_{i=1}^{1} \sum_{i=1}^{1} (\alpha_i^* - \alpha_i)(\alpha_j^* - \alpha_j)(x_i \cdot x_j) \tag{3}$$

By maximizing the above function under the following conditions, the coefficients $\alpha$ and $\alpha^*$ are obtained. These coefficients are called Lagrange coefficients.

$$\begin{cases} \sum \alpha_i^* = \sum \alpha_i \\ 0 \leq \alpha_i^* \leq C \qquad \text{for } i = 1, 2, \cdots, 1 \\ 0 \leq \alpha_i \leq C \end{cases} \tag{4}$$

In Equation (4), L represents the Lagrange function and C represents the penalty parameter or adjustment parameter. Notably, the aforementioned optimization issue may be handled using quadratic programming (QP) techniques. As a consequence, attaining the absolute maximum or minimum will be deterministic, and there is a chance of being caught in a local maximum or minimum. The final response will be arranged as follows:

$$w_0 = \sum_{supportvectors} (\alpha_i^* - \alpha_i) x_i$$
$$b_0 = -(1/2) w_0 \cdot [x_r + x_s] \tag{5}$$
$$f(x) = \sum_{supportvectors} (\alpha_i^* - \alpha_i)(x_i - x) + b_0$$

In this equation, $x_i$ is the input vector with which the model is trained, $x$ is the input vector, $x_r$ and $x_s$ are two vectors, $w_0$ supports the optimal weight vector, and $b_0$ is the optimal bias value. Data whose corresponding Lagrange coefficients are non-zero are known as support vectors. From the point of view of geometry, these data have a bigger prediction error than $\pm \varepsilon$, so the support vectors are not included in the range of $\pm \varepsilon$ and

it controls the number of support vectors. According to the relationship, it is observed that the data whose Lagrange coefficient are zero do not play a role in the final answer; in other words, they are support vectors that determine the final regression function with the optimal answer [24].

To build the support vector machine model, parameters C and $\varepsilon$ are defined by the user. The parameter C is an adjustment parameter and can accept values from zero to infinity. When large values are assigned to this parameter, SVM does not allow errors to occur in the training data and the result will be a complex model, so the generalizability of the model decreases. On the other hand, when C tends to zero, the model can accept a large error; as a result, the complexity of the model will be less [24].

The parameter $\varepsilon$ can also accept values from zero to infinity. The value of this parameter is very effective in the case of support vectors and as a result the efficiency of the model. Although the selection of very large values of $\varepsilon$ causes a reduction in the number of support vectors, which is also desirable, reaching this goal by widening the interval is incorrect. On the other hand, very small values of this parameter require a large number of support vectors to be selected and the possibility of overfitting increases.

The problem of linear regression in SVM can be easily extended to nonlinear regression. For this purpose, kernel functions are used. Kernel functions map the data to a feature space in which it is possible to use linear regression. So far, various kernel functions have been introduced, among which they can be classified as polynomial kernels. The radial basis function has been pointed out [25,26].

*2.4. Ant Colony System (ACS)*

This method, one of the innovative algorithms based on collective intelligence and population, is introduced as ant colony optimization [28]. The optimization of the ant colony, including a set of methods based on the behavior of the ant colony, is inspired by the search for food. To find food, ants do not follow the same route, which is the shortest possible route. Each ant emits a chemical substance called a pheromone along the way. All members of the colony sense the substance and direct their movement to the path that has more pheromone. In other words, the path with more pheromone will be more attractive to be chosen by an ant [29]. The ACS algorithm is one of the most efficient versions of the ants' algorithm, and is currently used in various practical research problems. The purpose of the ACS algorithm is to determine the number of ants in a graph that corresponds to the optimal problem. Each ant can move in this graph in a possible way, and based on the amount of pheromone and innovative actions, it can respond. Then, the ants will determine the quantity of routes based on the quality of the generated answer, and in this way, the communication between the ants will be improved. The ants are a special kind of natural ants that have innovative obedience and memory to record their previous movements [30]. The heuristics are defined based on the objective function of the problem in such a way that it indicates the degree of improvement in the value of this function due to the movement of an ant from one node to another node. In addition, every movement that an artificial ant can make is stored in a memory, so that it can easily go back and restore the pheromone values. In addition, due to the complex structure of the problem, only a few applications of the ants' algorithm have been used to solve the problem. This article presents the ACS method to solve the problem under study, which will be discussed in sufficient detail in the next sections. So far, different versions of the ACO algorithm have been presented [31]. Among these algorithms are the Ant Colony System Algorithm (AS), Max–Min Ant System (MMAS), and ACS. The main difference between the parts of the ACO algorithm is in the method of updating their pheromones [32]. The ACS algorithm is one of the most efficient ACO algorithms [33]. The main parameters using the algorithm are in Abbreviations.

Note that choosing $q = 0$ converts the ACS method to the AS method. Therefore, when and is less than equal, the ants use exploration to select the task as the next task in the schedule, while if it is greater, the ants use probability-based extraction to select the next task.

In this step, $k$ numbers of ants are randomly assigned to one of the tasks of the problem. These tasks are the starting point for each ant to build its answer. We assign the value of $\tau_0$ to the power of the value of the first pheromone to all the arcs of the problem ($\forall (i,j)\tau_{ij} = \tau_0$) and the amount of bow pheromone is $(i,j)$. Usually, the initial amount of pheromone is considered $\tau_0 = \frac{1}{(n \times \text{HA\_Value})}$ where HA_Value is the cost obtained by the heuristic algorithm.

In the solution-construction phase, each ant constructs a sequence in $n$ steps. In each step, ant $k$, which is in the $i$th task, calculates $j$ with the probability $q_0$ of its next task according to the Equation (6).

$$j = \arg \max_{l \in \mathbb{N}_i^k}\{[\tau_{il}]^\alpha [\eta_{il}]^\beta\} \tag{6}$$

The ant considers the probability $1 - q_0$ of its next task according to the Equation (7).

$$p_{ij}{}^k = \frac{[\tau_{ij}]^\alpha [\eta_{ij}]^\beta}{\sum_{l \in_i^k}[\tau_{il}]^\alpha [\eta_{il}]^\beta}, \text{if } j \in \mathbb{N}^k \tag{7}$$

where $d_{ij}$ is the length of the bow $(i, j)$, $\eta_{ij} = \frac{1}{d_{ij}}$ represents the influence of this innovative value, and $\beta$ is the innovative value. $\mathbb{N}^k$ is the set of candidate actions of the ant for its next move.

After task j is selected for the next move of ant $k$, it is removed from $\mathbb{N}^k$. In other words, until the end of the solution-creation process, ant $k$ no longer has the right to choose the work. It should also be said that at the beginning of the solution-creation process, $\mathbb{N}^k = V\ 1 \le k \le K$, where V is the set of all tasks of the problem.

Local updating of pheromones occurs during the solution-construction process. As soon as the bad ant passes an arc $(i, j)$, the amount of pheromone on that arc is also updated according to Equation (8). This process is called local updating of pheromones.

$$\tau_{ij} = (1 - \rho)\tau_{ij} + \rho\tau_0 \qquad 0 < \rho < 1 \tag{8}$$

When the process of building the answer for all ants is completed and the local search process is completed on all these answers, the global update process of pheromones is performed. In the global update of pheromones, only the root pheromone amount of the best answer obtained in each iteration, i.e., $T^{bs}$, is updated according to Equation (9).

$$\tau_{ij} = (1 - \rho)\tau_{ij} + \rho\Delta\tau_{ij}^{bs}, \forall (i, j) \in T^{bs} \tag{9}$$

where $\rho$ is the evaporation rate, $\Delta\tau_{ij}^{bs} = \frac{1}{C^{bs}}$, and $C^{bs}$ is the value of $T^{bs}$. Of course, these actions encourage other ants to repeat this tour.

The ACS algorithm is terminated if the number of repetitions of the algorithm reaches its maximum value, determined in the parameter setting. The flowchart of the study is presented in Figure 4.

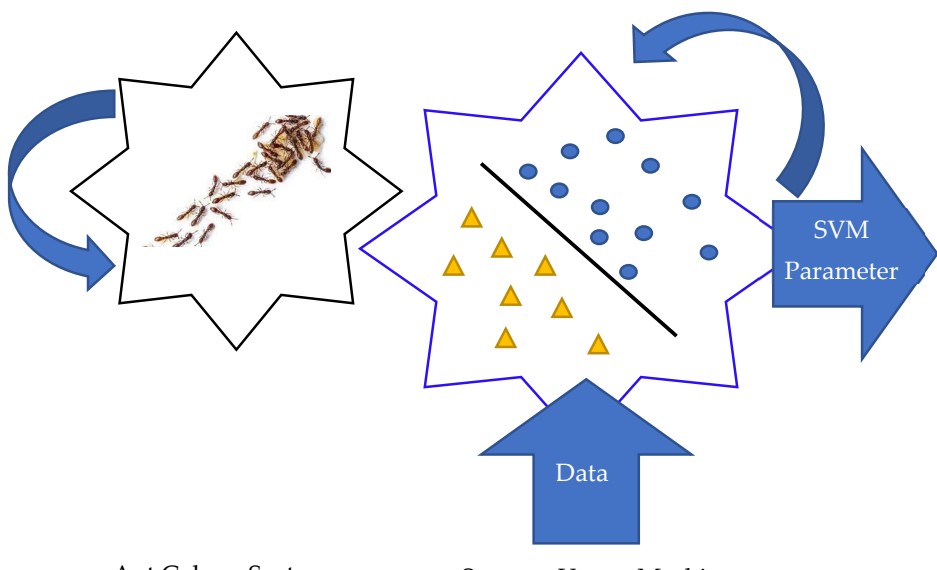

Ant Colony System          Support Vector Machine

**Figure 4.** An overview of the ACS and SVM model architecture.

## 3. Results and Discussion

The number of test data is equivalent to 150, whereas the amount of training data is equal to 600. In this particular investigation, the characteristics of the wall, roof, and glass as well as the ideal percentage of window to wall in each of the four directions make up the inputs of the network. The weights and biases are the values that are assigned to the inputs and the amount of energy consumption and $CO_2$ makes up the output of the model. In other words, this research involved the training of two different networks: one network was trained to collect energy, while the second network was trained to obtain $CO_2$. Because training data and test data are both utilized in the process of training the network, the results that the network produces for these two distinct types of data have been analyzed.

The evaluation results of each of the models are evaluated for changes in $\varepsilon$ values while other parameters are fixed, using the number of support vectors on the training set and the values of the correlation coefficient and the root mean square on the test data set. As $\varepsilon$ increases, the number of support vectors decreases. In the process of reaching the optimal response, reducing the number of the support vector by increasing the value is effective up to a certain level, and reaching the least error in the prediction and the most correlation with the actual values is not necessary. By examining the extended models with the radial basis kernel function, it is observed that the model in which the width of the kernel function is equal to 1.5 and the $\varepsilon$ is selected 0.007.

Figures 5a and 6a illustrate, respectively, the amount of energy consumed and the amount of $CO_2$ produced by trains. The precision of the network's training is demonstrated by the fact that the amount of energy that is computed by the network is very near to its actual value for each of the data points. Figures 5b and 6b, meanwhile, illustrate the statistics on the use of energy and the production of $CO_2$, respectively, for the tests.

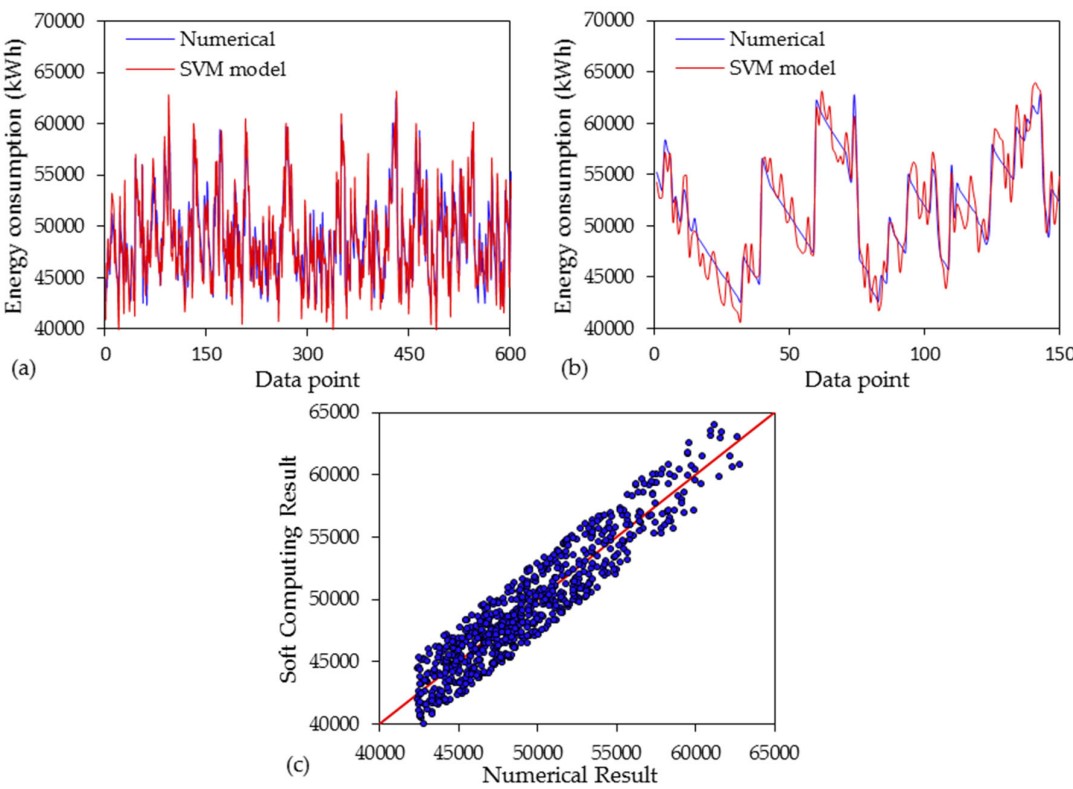

**Figure 5.** Comparison of the SVM-model-predicted energy consumption for (**a**) training data and (**b**) test data, and (**c**) scatter plot of the SVM and numerical results.

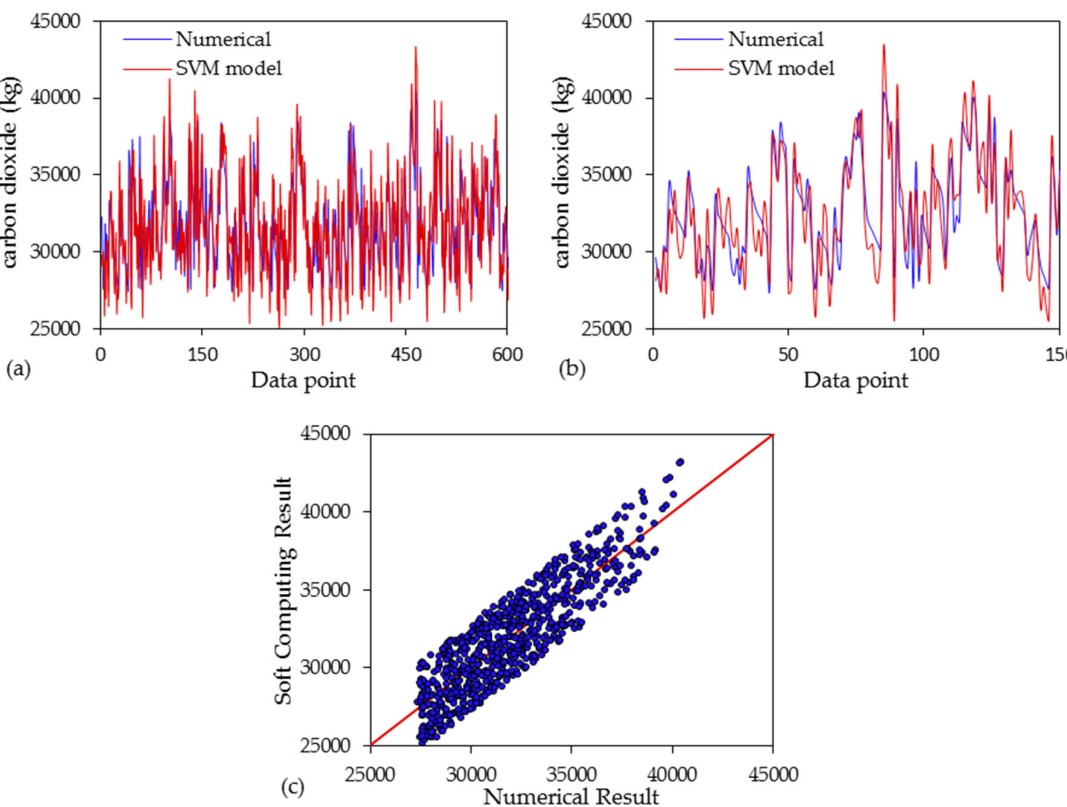

**Figure 6.** Comparison of the SVM-model-predicted $CO_2$ production for (**a**) training data and (**b**) test data, and (**c**) scatter plot of the SVM and numerical results.

The scatter plot of the SVM model and the simulated results are shown in Figures 5c and 6c, respectively. The comparison of the actual data with the data obtained from the network is shown in these figures, along with the best line that can be drawn through these points.

In general, in Figures 5 and 6, the closeness of the modeled and estimated values can be seen well, and this indicates the correctness of the SVM model and the sameness of the results obtained from the network and what was actually performed. The statistical index and the performance of the model are presented in Table 3.

**Table 3.** Statistical performance of the SVR models.

| Output | Stage | Statistical Index | | |
|---|---|---|---|---|
| | | R | $R^2$ | RMSE |
| Energy consumption | Train | 0.921 | 0.874 | 904 (kWh) |
| | Test | 0.882 | 0.824 | 1012 (kWh) |
| $CO_2$ | Train | 0.901 | 0.831 | 1129 (kg) |
| | Test | 0.863 | 0.799 | 1076 (kg) |

The results of Hui et al. [34] showed that depending on the geographical region, season, and direction of the building, it is possible to increase the temperature of the outer and inner surface to 27 and 1.0 degrees centigrade, respectively, by using different coatings. In the current research, insulation with a thickness of 5 cm was used in the wall and ceiling, which caused temperature and energy regulation. Therefore, the role of insulation and various coatings is important to reduce energy consumption.

## 4. Results of Optimization

As seen in the training of the network, the number of seven inputs includes the same input variables in the simulation. The number of outputs is two outputs, whose purpose is to minimize the amount of energy and $CO_2$. Therefore, ACS are executed once for energy consumption and once for $CO_2$, and the results are analyzed. The implementation of the algorithm shows that for both objective functions, the input values are almost the same.

Before implementing ACS to solve the problems, the algorithm parameters must be adjusted first. In this way, the most important parameters of the proposed algorithm, which have a significant effect on the convergence of the algorithm, are analyzed. For this purpose, different values are tested for each desired parameter, so that the best values for the investigated parameters are obtained in the selection process. In order to select quantitative variables, an experiment was designed according to the table, and for each number of tasks in each group, problems were randomly generated. Since the values of the parameters of the problem affect its current time, in this paper, different values are considered for a number of important parameters of the problem. This will show how the values of the parameters will affect the time to solve the problems. Parameter setting results for the method are shown in Table 4.

**Table 4.** Parameter setting results for the method.

| max_it | $q_0$ | | | $\beta$ | | | $\alpha$ | | $\rho$ | | | $k$ | | |
|---|---|---|---|---|---|---|---|---|---|---|---|---|---|---|
| 120 | 0.6 | 0.79 | 0.98 | 1 | 3 | 5 | 1 | 2 | 0.1 | 0.5 | 0.9 | 2 | 5 | 8 |

The optimal values of the input parameters resulting from the implementation of ACS for the objective function of minimizing the amount of energy consumption and $CO_2$ production are given. The optimal energy consumption is equal to $3.512 \times 10^4$ kWh and the amount of $CO_2$ production is equal to $2.151 \times 10^4$ kg (Figure 7).

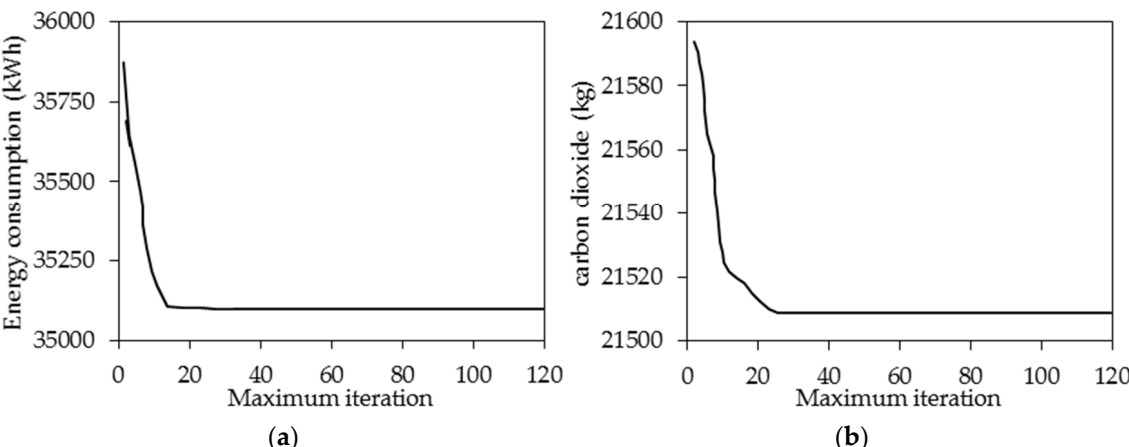

**Figure 7.** The results of (**a**) energy consumption and (**b**) $CO_2$ production variation in terms of 120 maximum iterations.

The optimal values of parameters include brick walls with insulation with a thickness of 40 mm, roof made of beams with insulation with a thickness of 40 mm, and triple-paned glass. According to the findings, the proportion of the north and east windows to the wall in one direction is 70 percent, and the proportion of the south window to the wall in that direction ranges between 35 and 50 percent. The amount of energy consumption and $CO_2$ dioxide that is produced is reduced to a negligible level when the ratio and percentage of the west window to the west wall is between 60 and 70 percent.

## 5. Conclusions

In this study, the current condition of an educational building in Jakarta, Indonesia was analyzed in terms of energy consumption and $CO_2$ emissions, and optimization was modeled using DesignBuilder, an SVM model, and an ACS algorithm. The application of ACS revealed that if the building includes a brick wall with 40 mm thick insulation, a beam roof with 40 mm thick insulation, three-pane glass, a proportion of north and east windows to the wall in the same direction of 60%, a proportion of the south window to the south wall between 10 and 14%, and a proportion of the west window to the west wall between 35 and 50%, the amount of energy and $CO_2$ is minimal.

Other effective methods for reducing energy consumption in buildings were investigated in this study, and the findings were generally associated with energy consumption optimization. This implies that by modifying the building's parameters, its energy consumption can be reduced. In addition, applying a large number of these variables does not incur significant costs in the building, and on the contrary, they decreased the building's current costs and the cost of energy. In this study, the parameters that lead to a reduction in energy consumption and $CO_2$ emissions are examined, including the optimal window-to-wall ratios in different building orientations. If the objective function of energy is chosen, the ACS results are in close agreement with the optimization results when the objective function is $CO_2$ concentration. In other words, the optimization of one objective function will result in the optimization of the other. Thus, we will not have to optimize two objectives for this specific problem. Consequently, the policy model for reducing greenhouse gases can also be presented. It is suggested that considering the climate conditions besides the building's physical features should be evaluate in the model performance in future studies.

**Author Contributions:** Conceptualization, W.A.; Methodology, W.A. and A.A.-M.; Software, I.H.A.-K. and Y.C.-E.; Validation, I.H.A.-K. and Y.C.-E.; Formal analysis, S.A.A. and A.A.-M.; Investigation, I.M. and I.H.A.-K.; Resources, S.A.A.; Data curation, W.A., I.M. and A.A.-M.; Writing—original draft, A.A.-M.; Writing—review and editing, Y.C.-E.; Visualization, Y.C.-E. All authors have read and agreed to the published version of the manuscript.

**Funding:** This research received no external funding.

**Institutional Review Board Statement:** Not applicable.

**Informed Consent Statement:** Not applicable.

**Data Availability Statement:** Not applicable.

**Conflicts of Interest:** The authors declare no conflict of interest.

## Abbreviations

| | |
|---|---|
| $\tau_{ij}$ | The pheromone value is on the edge that connects nodes *i* and *j*. |
| $p_{ij}^k$ | Probability of moving from node *i* to unvisited node *j* by ant *k*. |
| $\eta_{ij}$ | Innovative information to measure the ant's field of view |
| $\alpha,\ \beta$ | Parameters are controls that determine the importance ratio of the value of the ant's field of view against the pheromone marker on the edge connecting node *i* and *j*. |
| $q$ | A random parameter uniformly distributed in [0, 1]. |
| $q_0$ | A constant threshold parameter in [0, 1] that determines the importance ratio of mining to exploration. |

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
