# Peer review of "Energy Consumption and Carbon Dioxide Production Optimization in an Educational Building Using the Supported Vector Machine and Ant Colony System"

_sustainability, doi:10.3390/su15043118_

Round 1

Reviewer 1 Report

1.     Section 2.1: the authors provided detailed information about the location where the target building is located. However, I don’t think lines 89-94 are necessary since the study focuses on the single building, not the city.

2.     Section 3: the title of the section is “results and discussion.” While the authors put a lengthy description of the target building and its modeling in the section. I suggest adding one more “case study” section to summarize the building’s information and how it was modeled. And also combine the information from section 2.1.

3.     Section 3: this section is wordy. Please remove unnecessary words once they are summarized in tables to increase readers’ efficiency.

4.     Section 3 line 273: specific details of the training and test data, such as the temporal granularity.

5.     Section 3 line 276: the authors mentioned that “the objective functions are the amount of energy and carbon dioxide.” Does it mean the sum of energy and carbon dioxide? Was the object function applied to two models? Make it more explicit, please.

6.     Section 3 lines 281-286: what’s the real data? Is it the data generated from the designbuider model? Also confused about figure 3.

7.     Section 3 line 288: double check if it should be the amount of carbon dioxide instead of “the amount of energy calculated.”

8.     Section 3 lines 296-297: use R2 as an index instead of correlation since the high correlation doesn’t mean high model fitting’s accuracy.

9.     Section 3: provide results for the test data since it’s important to show the models’ performance.

10.  Section 3: the authors developed the model of the target building in desighbuilder. Did the author calibrate the model? What’s the calibration result?

11.  Section 3 line 269-270: did the authors assign different values to each window-to-wall ratio? If it did, is it reasonable? It’s a bit weird that the window-to-wall ratio is different for each wall.

12.  Section 3: how did the authors determine the hyperparameters of the SVM model? Report the process and the values, please.

13.  Section 4 line 301: the authors stated that the number of the outputs is 2, which should be energy consumption and carbon dioxide if I’m right. However, in lines 277-278, the authors mentioned that “two networks were trained in this research, one network to obtain 277 energy and one network to obtain carbon dioxide.” I’m confused.

14.  Section 5 line 34: discuss the reference in the discussion section instead of the conclusion.

15.  Honestly, several studies show that model parameters such as equipment load, lighting load, ventilation, or infiltration are more significant than envelopes. I cannot find any solid evidence to support the significance of only mapping envelopes to energy/CO2.

Author Response

Thank you for reviewing. Please find the response in the attachment.

Reviewer 2 Report

Although the research subject and objective are relevant to the field of energy efficient buildings and reduced emissions, the work done including the methodology that is applied, are not original. It Is not clear for example the originality of the work conducted by the authors, and how they contribute to the state of the art. There is large literature on the subject and lot of work has been already done on similar subjects, it is not however clear how this work compares to the others, as well as how it differs from other work.

Author Response

Thank you for your comments. Please find the response in the attachment.

Reviewer 3 Report

Good work. Following corrections are suggested. 

1. Contributions and novelty should be added explicitly in introduction.

2. A flowchart of the algorithm is required before the results section. 

3. A clear comparison with very relevant approaches is required to demonstrate the proposed methodology. 

4. Formatting needs careful review. 

5. Future extension of work should be added in the conclusion. No need to cite a reference in conclusion. 

6. Some English language checks are required. 

7. Recent literature review is lacking. More recent literature review is strongly recommended to be addd. 

Author Response

(The authors gave the same response as above.)

Reviewer 4 Report

Following comments recommended improving the manuscript entitled “Energy Consumption and Carbon Dioxide Production Optimization in an Educational Building Using the Supported Vector Machine and Ant Colony System”:

1.      The novelty of the study's originality needs to be mentioned in the abstract. This version does not make it abundantly apparent.

2.      More quantitative results should be mentioned in the abstract than qualitative results.

3.      The authors should state why the study is critical. Why did they choose an educational building? Is there any critical issue in these buildings regarding “carbon dioxide production”? All of these fundamentals should be clarified in the abstract and introduction.

4.      The study's literature review needs to provide the chicken and egg relationship of the variables under focus. The literature should also be discussed to give the established gaps of the study and these should be modeled accordingly. Wide arguments should be provided on the subject matter and there should be a good and coherent flow and interlinkages of ideas within the structural composition of the literature review. 

5.      Discussion about results was so simple. It needs to revise extensively. It was not clear the objective of the manuscript.

6. To give our readers a sense of continuity, I can encourage you to identify journal publications of similar research in your papers. You should make a literature check of the papers published in recent years (2020, 2021, and even 2022) and relate the content of relevant papers to the results and findings presented in your publication.

7.       Why did the author use the SVM model? Which criteria encourage the author to use ACS? It should be clarified for the readers.

8.       Conclusion section was written poorly. It has to be expressed in a more detailed manner, with the primary emphasis being placed on the primary findings of the study. Suggestions could be included in the last paragraph of the Conclusion.

Author Response

(The authors gave the same response as above.)

Reviewer 5 Report

My opinion is that the paper may possibly be published in this journal if some adjustments were made to it. It is possible to improve the manuscript by taking into consideration the following comments:

1.      It is essential that the novel aspect of the study's originality be addressed throughout the entire manuscript. The abstract and introduction sections did not adequately convey this information to the reader.

2.      Abstract must be rewritten. It is not informatics for readers in this style.  

3.      The use of a spell checker is obligatory in the English language and style. 

4. In introduction section the related work can be added like:

A novel developed method to study the energy/exergy flows of buildings compared to the traditional method

Carbon dioxide emissions prediction of five Middle Eastern countries using artificial neural networks

5.      “Study Area” section has not enough information about building structures, climate information, etc. All parameters used in the simulation should state in the manuscript.

6.       Line 226 to 233 “Studies related to electricity bills show that the highest electricity consumption is in the hot months of the year (June and July), the highest consumption in the middle load periods is 213126-kilowatt hours in a year, the peak electricity load is 409555-kilowatt hours and the lowest load is the electricity demand was 49868-kilowatt hours.

In the existing state, the wall material is brick type, the roof material is beam type, the cooling system is of split type, the window is made of wood with double-glazed glass, the mentioned information was entered in Design Builder software, and the heat transfer coefficient for each of the components of the building was calculated. In order to simulate and enter the details, the information in the software has been used.” This paragraph is unclear. I am not sure what you want to say.

7.      Is the SVM model inputs observed or numerical? If the data is observed to represent the data source, if the input data is numerical, the model and software validation must be stated.   

8.      Figure 1 is general! This figure should clarify in detail.

9.      I suggested that the outcomes be compared with other studies' results in this field.

10.      The description of the figures can be more detailed than the present.

Author Response

(The authors gave the same response as above.)

Round 2

Reviewer 1 Report

The authors addressed my comments well. I have no more comments.

Reviewer 2 Report

The authors have responded to the earlier comments.

Reviewer 5 Report

The paper is well revised now